# The Sampling Distribution of the Total Correlation for Multivariate Gaussian Random Variables

**DOI:** 10.3390/e21100921

**Published:** 2019-09-22

**Authors:** Taylor Rowe, Troy Day

**Affiliations:** Department of Mathematics and Statistics, Queen’s University, Kingston, ON K7L 3N6, Canada; tlrowe1005@gmail.com

**Keywords:** mutual information, total correlation, multiinformation, sampling distribution, central limit theorem, multivariate mutual information

## Abstract

The sampling distribution of the total correlation (TC) for a *d*-dimensional standardized multivariate Gaussian random variable with an identity covariance matrix is derived. It is shown to be the distribution of a sum of generalized beta random variables. It is also shown that, for large dimension and sample size, a central limit theorem holds, providing a Gaussian approximation to the sampling distribution for high dimensional data.

## 1. Introduction

Mutual information quantifies the information shared between two random variables [1,2,3]. This concept can be been generalized to *d* variables in a variety of ways [4,5,6,7], with the most direct generalization being Watanabe’s total correlation (TC),
(1)$$T\left(X\right)\equiv \sum_{i=1}^{d}h\left(X_{i}\right)-h\left(X\right)$$
where X is a vector whose components are the *d* random variables $X_{1},\ldots ,X_{d}$, and for continuous random variables, $h(X_{i})$ is the differential entropy of $X_{i}$ and h(X) is the joint differential entropy of X.

Total correlation is also sometimes called multivariate mutual information, and it is the Kullback–Leibler divergence between the joint density of X and the density obtained by taking the product of the marginal densities of the $X_{i}$. Thus, the total correlation T(X) quantifies, in a quite general sense, the information shared among all the *d* random variables. The total correlation is non-negative and in the case where all *d* random variables are mutually independent we have T(X)=0 [7,8]. For the special case where X is multivariate Gaussian with arbitrary mean and covariance matrix Σ, the total correlation can be written explicitly as
(2)$$\begin{array}{cc} T(X) & =\frac{1}{2}\sum_{i=1}^{d}\log\sigma_{ii}^{2}-\frac{1}{2}\log\left|\Sigma \right| \end{array}$$
where $\sigma_{ij}^{2}$ is the ijth entry of Σ. When the $X_{i}$ are independent we have $\sigma_{ij}^{2}=0$ for all i≠j and so $\log\left|\Sigma \right|=\log\sigma_{11}^{2}\sigma_{22}^{2}\cdots \sigma_{dd}^{2}$, giving T(X)=0 in Equation (Equation 2) as expected.

The total correlation provides a natural way to quantify dependencies among a set of random variables. For example, often we seek to determine if a set of random variables are mutually independent because dependency among variables can indicate interesting and meaningful relationships in nature. To do so one can take a sample from the unknown distribution and compute the total correlation from this sample. Even if the random variables *are* mutually independent, however, the total correlation measured using such a finite sample will typically be positive (rather than zero) simply because of sampling variation. Therefore, it is of interest to know the sampling distribution of the total correlation under independence. Once we have the sampling distribution we can then perform statistical tests of independence. Here we derive the sampling distribution of (Equation 2) in the case where the $X_{i}$ are standardized (i.e., zero mean, unit variance), independent, Gaussian random variables.

Previous authors have proposed exact expressions for the mean and variance of the sample total correlation [9,10]. In fact, Guerrero (Section 2.1 of [9]) derived a moment generating function for the sample total correlation using the distribution of the log-determinant of a Wishart matrix (see Wilks [11,12]). Unfortunately the asymptotic approximation of Guerrero’s result does not match the results of Marralec [10] suggesting that one of the two is incorrect. We will resolve this discrepancy by deriving the moment generating function directly from our expression for the probability density function of the sample total correlation. In the limit of large sample size our results match those presented in Section 4.1 of Marralec [10], suggesting that the moment generating function of [9] is incorrect.

## 2. Definitions and Preliminaries

Let X represent a *d*-variate zero mean Gaussian random variable with covariance matrix $\Sigma =I_{d}$ where $I_{d}$ is the *d*-dimensional identity matrix. Let $\left\{x_{1},\cdots ,x_{n}\right\}$ denote a sample of *n* draws from the distribution of X. We focus on the case where n≥d. The sample covariance matrix is $\hat{\Sigma }=\left(1/n\right)\sum_{i=1}^{n}x_{i}x_{i}^{\prime }=\left\{{\hat{\sigma }}_{ij}\right\}$ and $n\hat{\Sigma }$ is Wishart distributed with *n* degrees of freedom, which we denote as $n\hat{\Sigma }∼W\left(\Sigma ,d,n\right)$. From Equation (Equation 2) the sample total correlation is then also a random variable and is computed as
(3)$$\begin{array}{cc} {\hat{T}}_{d,n}\left(X\right) & =\frac{1}{2}\sum_{i=1}^{d}\log{\hat{\sigma_{ii}}}^{2}-\frac{1}{2}\log|\hat{\Sigma }| \end{array}$$
where the subscripts *d* and *n* indicate that $\hat{T}$ is a family of random variables indexed by dimension and sample size.

Odell and Feiveson’s 1966 result [13] provides a convenient way to characterize a Wishart-distributed matrix. Suppose that $V_{i}^{\left(n\right)}$(1≤i≤d) are independent chi-square random variables with n−i+1 degrees of freedom. Suppose that $N_{ij}$ are independent standardized normal random variables for 1≤i<j≤d, also independent of every $V_{i}^{\left(n\right)}$. Now construct the random variables
(4)$$\begin{array}{cc} b_{11} & =V_{1}^{\left(n\right)}\\ b_{jj} & =V_{j}^{\left(n\right)}+\sum_{i=1}^{j-1}N_{ij}^{2},\;\;\;\;\;\;\;2\leq j\leq d\\ b_{1j} & =N_{1j}\sqrt{V_{1}^{\left(n\right)}}\;\;\;\;\;\;\;\;\;2\leq j\leq d\\ b_{ij} & =N_{ij}\sqrt{V_{i}^{\left(n\right)}}+\sum_{k=1}^{i-1}N_{ki}N_{kj},\;\;\;2<i<j\leq d. \end{array}$$

Then the matrix $B=\{b_{ij}\}$ (with $b_{ij}=b_{ji}$) is Wishart-distributed $W(I_{d},d,n)$ and thus we have
(5)$$\begin{array}{cc} n{\hat{\sigma }}_{ii}^{2}∼b_{ii} & ∼V_{i}^{\left(n\right)}+A_{i}\;\;1<1\leq d \end{array}$$
where $A_{i}$ are independent chi-square random variables with i−1 degrees of freedom and we define $A_{1}=0$. Now following [14] we can also define the lower-triangular matrix $T=\{t_{ij}\}$ as
(6)$$\begin{array}{cc} t_{ii} & =\sqrt{V_{i}^{\left(n\right)}}\\ t_{ij} & =N_{ji}\;\;\;\;1\leq j<i\leq d\\ t_{ij} & =0\;\;\;\;\;i<j\leq d \end{array}$$
and thus $B=TT^{\prime }$. Furthermore, $|B|=|TT^{\prime }{\left|=|T\right|}^{2}=\prod_{i=1}^{d}t_{ii}^{2}=\prod_{i=1}^{d}V_{i}^{\left(n\right)}$, revealing that
(7)$$n^{d}\left|\hat{\Sigma }\right|∼\prod_{i=1}^{d}V_{i}^{\left(n\right)}.$$

Result (Equation 7) is a special case of results found in Wilks [11]. For analogous results involving complex matrices see Goodman [15].

## 3. The Sampling Distribution of the Total Correlation

With the above preliminaries the we can now state the following theorem.

**Theorem** **1**(The Sampling Distribution of TC)**.**
*Consider a sample of size n from a set of d independent, standardized, Gaussian random variables, with n≥d. The total correlation (TC) is distributed as*
(8)$$\begin{array}{cc} {\hat{T}}_{d,n}\left(X\right) & ∼\frac{1}{2}\sum_{i=1}^{d-1}\log\left(1+\frac{i}{n-i}F_{i,n-i}\right) \end{array}$$
*where $F_{i,n-i}$ are independent F-distributed random variables with i and n−i degrees of freedom. Equivalently, (Equation 8) can be written as*
(9)$$\begin{array}{cc} {\hat{T}}_{d,n}\left(X\right) & ∼\sum_{i=1}^{d-1}Y_{i,n} \end{array}$$
*where $Y_{i,n}$ is a beta-exponential random variable with probability density*
$$f_{Y_{i,n}}\left(y\right)=\lambda \frac{{\left(1-e^{-\lambda y}\right)}^{\frac{i}{2}-1}{\left(e^{-\lambda y}\right)}^{\frac{n-i}{2}}}{B(\frac{i}{2},\frac{n-i}{2})}\;\;\;y>0$$
*having parameter λ=2.*

**Proof.** Writing Equation (Equation 3) as
(10)$$\begin{array}{cc} {\hat{T}}_{d,n}\left(X\right) & =\frac{1}{2}\log\frac{\prod_{i=1}^{d}{\hat{\sigma_{ii}}}^{2}}{|\hat{\Sigma }|} \end{array}$$
and using result (Equation 5) and (Equation 7) one obtains
$$\begin{array}{cc} {\hat{T}}_{d,n}\left(X\right)\; & ∼\frac{1}{2}\log\frac{\prod_{i=1}^{d}\left(V_{i}^{\left(n\right)}+A_{i}\right)}{\prod_{i=1}^{d}V_{i}^{\left(n\right)}}\\ \; & ∼\frac{1}{2}\log\prod_{i=1}^{d}\left(1+\frac{A_{i}}{V_{i}^{\left(n\right)}}\right)\\ \; & ∼\frac{1}{2}\sum_{i=1}^{d}\log\left(1+\frac{A_{i}}{V_{i}^{\left(n\right)}}\right). \end{array}$$Scaling each chi-square by their corresponding degrees of freedom and re-indexing, yields (Equation 8). Equivalently, if we define $Y_{i,n}=\frac{1}{2}\log\left(1+\frac{i}{n-i}F_{i,n-i}\right)$ then ${\hat{T}}_{d,n}\left(X\right)∼\sum_{i=1}^{d-1}Y_{i,n}$, and using standard techniques it be can shown that the random variable $Y_{i,n}$ has probability density
$$f_{Y_{i,n}}\left(y\right)=2\frac{{\left(1-e^{-2y}\right)}^{\frac{i}{2}-1}{\left(e^{-2y}\right)}^{\frac{n-i}{2}}}{B(\frac{i}{2},\frac{n-i}{2})}\;\;\;y>0$$
where B(x,y) is the beta function. ☐

**Corollary** **1.**
*The moment generating function for ${\hat{T}}_{d,n}\left(X\right)$ is*
(11)$$\begin{array}{cc} M_{d,n}\left(t\right) & ={\left[\frac{\Gamma (\frac{n}{2})}{\Gamma (\frac{n-t}{2})}\right]}^{d-1}\;\prod_{i=1}^{d-1}\frac{\Gamma (\frac{n-i-t}{2})}{\Gamma (\frac{n-i}{2})} \end{array}$$
*where Γ(x) is the gamma function. The mean and variance of ${\hat{T}}_{d,n}\left(X\right)$ are therefore*
(12)$$\begin{array}{cc} \mu_{d,n} & =\frac{d-1}{2}\psi \left(n/2\right)-\frac{1}{2}\sum_{i=1}^{d-1}\psi \left(\frac{n-i}{2}\right)\\ \sigma_{d,n}^{2} & =-\frac{d-1}{4}\psi^{\left(1\right)}\left(n/2\right)+\frac{1}{4}\sum_{i=1}^{d-1}\psi^{\left(1\right)}\left(\frac{n-i}{2}\right) \end{array}$$
*where $\psi \left(x\right)=\Gamma^{\prime }\left(x\right)/\Gamma \left(x\right)$ is the digamma function and $\psi^{\left(k\right)}\left(x\right)$ denotes its $k^{th}$ derivative.*


**Proof.** Taking $Y_{i,n}=\frac{1}{2}\log\left(1+\frac{i}{n-i}F_{i,n-i}\right)$, the moment generating function for $Y_{i,n}$ is
$$\phi_{i,n}\left(t\right)=E\left[e^{tY_{i,n}}\right]=\frac{\Gamma \left[\frac{n}{2}\right]\Gamma \left[\frac{n-i-t}{2}\right]}{\Gamma \left[\frac{n-i}{2}\right]\Gamma \left[\frac{n-t}{2}\right]}.$$
The random variables in the sum $\sum_{i=1}^{d-1}Y_{i,n}$ are independent, and therefore the moment generating function $M_{d,n}\left(t\right)$ for ${\hat{T}}_{d,n}\left(X\right)$ is the appropriate product of the functions $\phi_{i,n}\left(t\right)$. Equation (Equation 12) then follow directly from the properties of moment generating functions. ☐

Guerrero [9] obtained a formula for the mean and variance of ${\hat{T}}_{d,n}\left(X\right)$ (except for a typo in the variance) using Wilks’ [12] moment generating function for the generalized variance. These are remarkably close to (Equation 12), but the proposed moment generating function for the sample total correlation information provided in Guerrero [9] appears to be incorrect.

## 4. A Central Limit Theorem for the Total Correlation

Girkos central limit theorem [16] implies asymptotic normality of the sample log-determinant, as seen in the work of Bao et al., and Cai et al. [17,18]. This suggests the existence of a central limit theorem result for ${\hat{T}}_{d,n}\left(X\right)$ when the dimension *d* and sample size *n* are large. Here we provide such a result.

Define the mean and variance of $Y_{i,n}$ as $m_{i,n}=E\left[Y_{i,n}\right]$ and $s_{i,n}^{2}=E\left[{\left(Y_{i,n}-\mu_{i,n}\right)}^{2}\right]$, and the mean-centered random variables $Y_{i,n}^{*}=Y_{i,n}-m_{i,n}$. Note that $\sigma_{d,n}^{2}=\sum_{i=1}^{d-1}s_{i,n}^{2}$.

**Theorem** **2**(Asymptotic normality of TC)**.**
*Suppose n→∞ and d→∞ in such a way that n/d→k where 1≤k<∞. Then*
(13)$$\frac{1}{\sigma_{d,n}^{2}}\sum_{i=1}^{d-1}Y_{i,n}^{*}\overset{}{\to }N\left(0,1\right)$$
*where convergence is in distribution. Thus, for large n and d (with n≥d) the total correlation ${\hat{T}}_{d,n}\left(X\right)$ is approximately normally distributed with mean and variance given by $\mu_{d,n}$ and $\sigma_{d,n}^{2}$ in Equations (Equation 12).*

**Proof.** The $Y_{i,n}^{*}$ are a triangular array of random variables such that, for any fixed *n* the $Y_{i,n}^{*}$ (1≤i≤d−1) are independent. Thus, (Equation 13) will hold provided that the Lyapunov condition is satisfied [19]; namely, that there exists a δ>0 such that
$$\lim_{d,n\to \infty }\frac{1}{\sigma_{d,n}^{2+\delta }}\sum_{i=1}^{d-1}E[|Y_{i,n}^{*}{|}^{2+\delta }]=0.$$For δ=2 the entries in Lyapunov’s summation represent each $Y_{i,n}$’s fourth central moment, for which the generating function is $C_{i,n}\left(t\right)=e^{-m_{i,n}t}\phi_{i,n}\left(t\right)$. The summation therefore becomes
$$\begin{array}{ccc} \sum_{i=1}^{d-1}E\left[{\left(Y_{i,n}^{*}\right)}^{4}\right] & = & \sum_{i=1}^{d-1}\frac{1}{16}\left(3{\left(\psi^{\left(1\right)}\left(\frac{n-i}{2}\right)-\psi^{\left(1\right)}\left(n/2\right)\right)}^{2}+\psi^{\left(3\right)}\left(\frac{n-i}{2}\right)-\psi^{\left(3\right)}\left(n/2\right)\right)\\ & = & \frac{3}{16}\sum_{i=1}^{d-1}{\left(\psi^{\left(1\right)}\left(\frac{n-i}{2}\right)-\psi^{\left(1\right)}\left(n/2\right)\right)}^{2}+\frac{1}{16}\sum_{i=1}^{d-1}\psi^{\left(3\right)}\left(\frac{n-i}{2}\right)-\frac{d-1}{16}\psi^{\left(3\right)}\left(n/2\right) \end{array}$$
while the denominator in Lyapunov’s condition is
$$\sigma_{d,n}^{4}={\left(\frac{1}{4}\sum_{i=1}^{d-1}\psi^{\left(1\right)}\left(\frac{n-i}{2}\right)-\frac{d-1}{4}\psi^{\left(1\right)}\left(n/2\right)\right)}^{2}.$$In Appendix A we show that
$$0\leq \frac{3}{16}\sum_{i=1}^{d-1}{\left(\psi^{\left(1\right)}\left(\frac{n-i}{2}\right)-\psi^{\left(1\right)}\left(n/2\right)\right)}^{2}+\frac{1}{16}\sum_{i=1}^{d-1}\psi^{\left(3\right)}\left(\frac{n-i}{2}\right)-\frac{d-1}{16}\psi^{\left(3\right)}\left(n/2\right)\leq \frac{48}{n-d+1}$$
and, for any fixed 1≤k<∞, and for sufficiently large *d* and *n* with n/d sufficiently close to *k*,
$$\frac{1}{4}{\left(\ln\left(\frac{n}{n-d+1}\right)+\frac{d-1}{n(n-d+1)}-\frac{d-1}{2}\left(\frac{2}{n}+\frac{4}{n^{2}}\right)\right)}^{2}\leq {\left(\frac{1}{4}\sum_{i=1}^{d-1}\psi^{\left(1\right)}\left(\frac{n-i}{2}\right)-\frac{d-1}{4}\psi^{\left(1\right)}\left(n/2\right)\right)}^{2}.$$Therefore, for any fixed 1≤k<∞, and for sufficiently large *d* and *n* with n/d sufficiently close to *k*, we have
(14)$$0\leq \frac{1}{\sigma_{d,n}^{4}}\sum_{i=1}^{d-1}E[|Y_{i,n}^{*}{|}^{4}]\leq \frac{\frac{48}{n-d+1}}{\frac{1}{4}{\left(\ln\left(\frac{n}{n-d+1}\right)+\frac{d-1}{n(n-d+1)}-\frac{d-1}{2}\left(\frac{2}{n}+\frac{4}{n^{2}}\right)\right)}^{2}}.$$Now first consider the case where n=d (and therefore k=1). Then (Equation 14) simplifies to
$$0\leq \frac{1}{\sigma_{d,n}^{4}}\sum_{i=1}^{d-1}E[|Y_{i,n}^{*}{|}^{4}]\leq \frac{48}{\frac{1}{4}{\left(\ln n+\frac{n-1}{n}-\frac{n-1}{2}\left(\frac{2}{n}+\frac{4}{n^{2}}\right)\right)}^{2}}.$$Taking the limit n→∞ yields zero on the right-hand side, verifying Lyapunov’s condition for k=1. Next, consider the case where n>d. Taking the limit in (Equation 14) as n→∞ and d→∞ in such a way that n/d→k where 1<k<∞, again we see that the right-hand side is zero. This verifies Lyapunov’s condition in the case where k>1, thereby completing the proof. ☐

## 5. Conclusions

The total correlation of a multivariate random variable (sometimes called multivariate mutual information) is the Kullback–Leibler divergence between the joint density of the random variable and the product of its marginal densities. It therefore provides a natural measure of the degree of independence of a set of random variables. In this paper we derived the sampling distribution of the total correlation for a *d*-dimensional standardized multivariate Gaussian random variable with identity covariance matrix, and showed that it is the distribution of a sum of generalized beta random variables. We also proved that, for large dimension and sample size, a central limit theorem holds, providing a Gaussian approximation to the sampling distribution for high dimensional data.

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
