# Peer review of "The Sampling Distribution of the Total Correlation for Multivariate Gaussian Random Variables"

_entropy, 2019, doi:10.3390/e21100921_

Round 1

Reviewer 1 Report

I like your paper.  I found this subject described only (except yours) in papers by Guerrero and Marrelec et al. and I think that it is important subject to be investigate also for other distributions.

I have only two small comments: it would probably be advisable to provide exact references to the formulas in paper by Guerrero and Marrelec et al. And could you write a few sentences why you think the subject you investigate is important.

Would it be possible to find the error in deriving the formula for moment generating function in the paper by Guerrero? However, this is  not necessary for publishing your work.

Author Response

Reviewer 1

I like your paper.  I found this subject described only (except yours) in papers by Guerrero and Marrelec et al. and I think that it is important subject to be investigate also for other distributions.

I have only two small comments: it would probably be advisable to provide exact references to the formulas in paper by Guerrero and Marrelec et al.

Done.

 And could you write a few sentences why you think the subject you investigate is important.

We now include some brief information about this in the introduction.

Would it be possible to find the error in deriving the formula for moment generating function in the paper by Guerrero? However, this is  not necessary for publishing your work.

We went back and tried to do this but unfortunately there are not enough steps of the derivation given to figure out where the error enters.

Reviewer 2 Report

The paper is well-written and, in principle, should be considered for publication.

The authors study the Gaussian limit $d\to\infty, n\to \infty, d/n\to$ const. This analysis is somewhat incomplete and may be extended to the case $d=n^{\alpha}, n\to \infty, 0\leq\alpha\leq 1$. In this case the sample Kullback-Leibler divergence multiplied by $n^{1-\alpha}$ should have a limit in distribution. I suggest that the authors resolve this question in the present publication to avoid straightforward generalizations in the future.

Author Response

The paper is well-written and, in principle, should be considered for publication.

The authors study the Gaussian limit $d\to\infty, n\to \infty, d/n\to$ const. This analysis is somewhat incomplete and may be extended to the case $d=n^{\alpha}, n\to \infty, 0\leq\alpha\leq 1$. In this case the sample Kullback-Leibler divergence multiplied by $n^{1-\alpha}$ should have a limit in distribution. I suggest that the authors resolve this question in the present publication to avoid straightforward generalizations in the future.

This sounds like it might be an interesting extension but we could not fully understand exactly what sort of result the reviewer had in mind or why. From our reading of the comment the result would be one in which the limiting distribution is not Gaussian. But if so, then we feel this sort of result is outside the primary goal of the paper. In particular, the goal was to provide the sampling distribution for the specific case of independent Gaussian RVs so that one could test for independence of RVs in this case. Our results provide a complete and exact description of this case, so from the perspective of the problem that motivated the manuscript, the analysis is complete. However, in the manuscript we also look at the limiting case of large n for two reasons. First, because previous authors have managed to obtain such asymptotic results and so this allows comparison. And second, because sometimes when n is large, even though one could still use the exact results we provide, there might be an approximation that gives a reasonably good solution to the problem but that is easier to use than the exact results. Cases where a CLT result holds such that, asymptotically, the sampling distribution is Gaussian are one such example because only a mean and variance are needed and there are then standard tools. We can see that one might want to examine other limits like the one suggested for other reasons but it was not clear to use how such results would be related to the goals of the manuscript. We are not sure if this adequately addresses the comment but we hope so.

Round 2

Reviewer 2 Report

I except that the paper has some merits in its present form.